# ADAPTIVE RETENTION & CORRECTION: TEST-TIME TRAINING FOR CONTINUAL LEARNING

**Haoran Chen**[1,2]  **Micah Goldblum**[3]  **Zuxuan Wu**[1,2†]  **Yu-Gang Jiang**[1,2]

[1]Shanghai Key Lab of Intell. Info. Processing, School of CS, Fudan University
[2]Shanghai Collaborative Innovation Center of Intelligent Visual Computing
[3]New York University

## ABSTRACT

Continual learning, also known as lifelong learning or incremental learning, refers to the process by which a model learns from a stream of incoming data over time. A common problem in continual learning is the classification layer's bias towards the most recent task. Traditionally, methods have relied on incorporating data from past tasks during training to mitigate this problem. However, the recent shift in continual learning to memory-free environments has rendered these approaches infeasible. In this study, we propose a solution focused on the testing phase. We first introduce a simple Out-of-Task Detection method, OTD, designed to accurately identify samples from past tasks during testing. Using OTD, we then propose: (1) an **Adaptive Retention** mechanism for dynamically tuning the classifier layer on past task data; (2) an **Adaptive Correction** mechanism for revising predictions when the model classifies the data from previous tasks into classes of the current task. We name our approach **Adaptive Retention & Correction** (ARC). Although designed for memory-free environments, ARC also proves effective in memory-based settings. Extensive experiments show that our proposed method can be plugged in to virtually any existing continual learning approach without requiring any modifications to its training procedure. Specifically, when integrated with state-of-the-art approaches, ARC achieves an average performance increase of 2.7% and 2.6% on the CIFAR-100 and Imagenet-R datasets, respectively. Code is available at Github Link.

## 1 INTRODUCTION

Deep learning has seen remarkable advances over the past decade (Krizhevsky et al., 2012; Ren et al., 2015; He et al., 2016; Ronneberger et al., 2015; Long et al., 2015; Dosovitskiy et al., 2021), largely due to the availability of large static datasets. However, real-life applications often require continually updating models on new data. A major challenge in this dynamic updating process is the phenomenon of catastrophic forgetting (Kirkpatrick et al., 2016; French, 1999), where a model's performance on previously learned tasks drastically degrades. To address this issue, various continual learning (CL) methods (De Lange et al., 2021; Hadsell et al., 2020; Mai et al., 2022) have been developed to provide a more sustainable and efficient approach to sequential model adaptation.

The community has identified two primary sources of catastrophic forgetting: 1) overfitting of the feature extraction network on new tasks (Li and Hoiem, 2017; Kirkpatrick et al., 2016; Rebuffi et al., 2017); and 2) bias towards new classes introduced by the linear classifier (Wu et al., 2019; Chrysakis and Moens, 2023; Belouadah and Popescu, 2019). Throughout the years, most works have been focusing on addressing forgetting of the representation layer. More recently, however, Zhang et al. (2023) show that for pretrained models, using a small learning rate for the representation layer and a larger learning rate for the classifier can effectively mitigate the overfitting issue, thereby relieving catastrophic forgetting. In this work, we take a step further and show that even for models overfitted on the latest task, the amount of information lost on the representation layer is minimal. As such, we primarily focus on tackling the latter issue, i.e., mitigating classifier bias.

---

† Corresponding author.

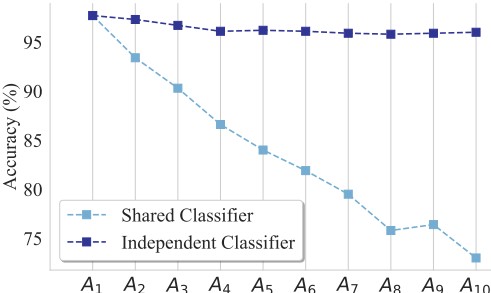

Figure 1: Toy example using pretrained ViT. Since the backbone network is fixed, the only cause of performance degradation is the bias introduced by the classifier.

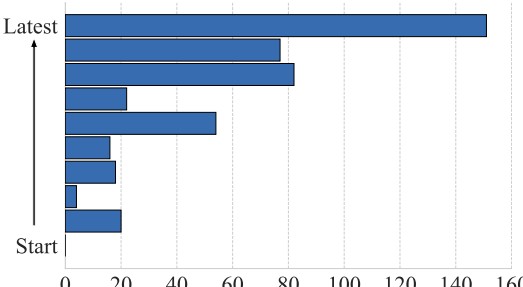

Figure 2: Distribution of incorrect prediction for first task samples after training on the latest task. As is shown, the predictions are skewed to the top, indicating the presence of classification bias.

First studied by Wu et al. (2019), classifier bias relates to the fact that the final classification layer tends to misclassify samples from previous tasks into classes from the current task due to imbalanced training data. To illustrate, we present a toy example in Figure 1, where we perform linear probing on Split Imagenet-R using a pretrained ViT (Dosovitskiy et al., 2021). The dark blue line in the graph shows the performance when an independent classifier is trained for each specific task, and the model selects the appropriate one for testing. The light blue line shows the performance of testing with only a single shared classifier that expands to accommodate new classes with each new task. As shown, the model experiences almost no performance drop when evaluated using independent classifiers, whereas there is a severe performance drop when using a shared classifier. To further demonstrate the cause of this performance degradation, we show in Figure 2 the distribution of incorrect predictions for data from the first task. It is evident that the classifier tends to favor classes of the current task, highlighting the inherent bias in the predictions of the shared classifier.

To address performance degradation in the classification head, previous methods (Wu et al., 2019; Belouadah and Popescu, 2019; Chrysakis and Moens, 2023) often rely on including an additional memory dataset when training. However, concerns over computational costs and privacy have prompted recent CL approaches (Wang et al., 2022d;c; Smith et al., 2023) to develop methods that operate without memory, making these memory-dependent methods less appealing. Nonetheless, we notice that an often-overlooked aspect of continual learning is the availability of data from previous tasks during inference. This availability offers a unique opportunity to retrospect prior knowledge without the need for any additional memory storage.

In practice, we often do not know what task test samples come from when they arrive. Therefore, our first step involves developing an effective method to correctly identify data from past tasks. Drawing inspiration from Out-of-Distribution (OOD) detection (Liang et al., 2017; DeVries and Taylor, 2018), we propose a simple Out-of-Task detection method, OTD, by leveraging the model's output confidence and predicted class. Specifically, OTD is built on top of the following two empirical findings:

1. Predicting that a test sample comes from a past task with high confidence generally indicates accurate classification of data from a previous task.

2. Predicting that a test sample comes from the current task with low confidence suggests a likely misclassification of data from a previous task.

With these two categories of predictions, we design two independent approaches tailored to each. Specifically, we propose **Adaptive Retention & Correction** (ARC) which consists of:

1. **Adaptive Retention**, where we utilize accurately classified samples from previous tasks and re-balance the classification layer in an online manner with one gradient update.

2. **Adaptive Correction**, where we correct predictions made on potentially misclassified samples from previous tasks by adjusting the classifier's preference for the most recent task's classes.

Given that ARC does not interfere with model training, we evaluate its compatibility with existing methodologies. To this end, we conduct extensive experiments on popular continual learning benchmarks by integrating ARC into both classical and state-of-the-art methods in the class-incremental setting, and show that ARC boosts their performance by a large margin. While designed for the memory-free scenario, our experiments show that ARC is equally effective using memory or not. Specifically, on the Split CIFAR-100 dataset, incorporating ARC can increase the average accuracy by 2.7%, with DER (Yan et al., 2021) showing the most significant improvement of 4.5%. On the Split Imagenet-R dataset, ARC can increase the average accuracy by 2.6%, with iCarL (Rebuffi et al., 2017) exhibiting the largest increase of 6.6%. In summary, our contributions are three-fold:

- We show that the strategic use of output confidence levels together with the output class can help distinguish between data from past tasks that is correctly classified and data from past tasks that is misclassified into classes from the current task, and propose OTD, a simple Out-of-Task detection method.

- We leverage the fact that models can interact with samples from past tasks during inference and propose Adaptive Retention & Correction (ARC), where the classifier weight is re-balanced and predictions on potentially misclassified images are rectified.

- Extensive experiments show that integrating ARC into both traditional and state-of-the-art methods produce significant improvements, with an average increase of 2.7% on Split CIFAR-100 and an average increase of 2.6% on Split Imagenet-R.

## 2 RELATED WORKS

### 2.1 CONTINUAL LEARNING

Continual Learning (CL) (Kirkpatrick et al., 2016; Li and Hoiem, 2017; Dhar et al., 2019; Hung et al., 2019; Douillard et al., 2022) has been an area of active research aiming to enable models to learn continuously from a stream of data while retaining previously acquired knowledge. A significant challenge in CL is catastrophic forgetting (French, 1999), where a model loses previously learned information upon learning new data. To address this notorious phenomenon, a plethora of approaches have focused specifically on tackling classifier bias. For example, BiC (Wu et al., 2019) introduces an additional bias correction layer that trains on a validation dataset composed of both data from old tasks and data from the current task. IL2M (Belouadah and Popescu, 2019) directly adjusts the logit values for classes of the current task using learned statistics. OBC (Chrysakis and Moens, 2023) proposes re-weighting the classifier layer based on the memory dataset.

Recently, the field of CL has witnessed a paradigm shift influenced by the exponential growth in model scales, where researchers are increasingly leveraging large pretrained models without any memory data (Wang et al., 2022c; Smith et al., 2023; Wang et al., 2022d; Chen et al., 2023). Within this context, prompt-based methods have gained significant attention. For example, L2P (Wang et al., 2022d) introduces a prompt pool from which task-specific prompts are dynamically selected. Dualprompt (Wang et al., 2022c) extends the idea by leveraging both task-specific and task-agnostic prompts. However, since traditional methods on tackling classifier bias mostly rely on leveraging data from the past task, methods like BiC can't operate under the current memory-free paradigm. To address this, a most recent work SLCA (Zhang et al., 2023) proposes to store the sample means and covariance matrix, and models the class feature as a Gaussian distribution. After full training on each task, the classifier is further balanced on features generated from it.

### 2.2 OUT OF DISTRIBUTION DETECTION

Out-of-distribution (OOD) detection (Yang et al., 2021; Salehi et al., 2021) is a technique used in machine learning to identify data points that differ significantly from the training data distribution. These data points, known as out-of-distribution samples, do not belong to any of the classes seen during training. OOD detection methods typically involve using a score function to measure the uncertainty or confidence of the model's predictions and flagging inputs with low confidence as out-of-distribution. For example, a maximum softmax probability (MSP) is used as the score function by Hendrycks and Gimpel (2016). Following this, the subsequently introduced ODIN (Liang et al., 2017) established a solid baseline method for the OOD detection community. Recently, due to their

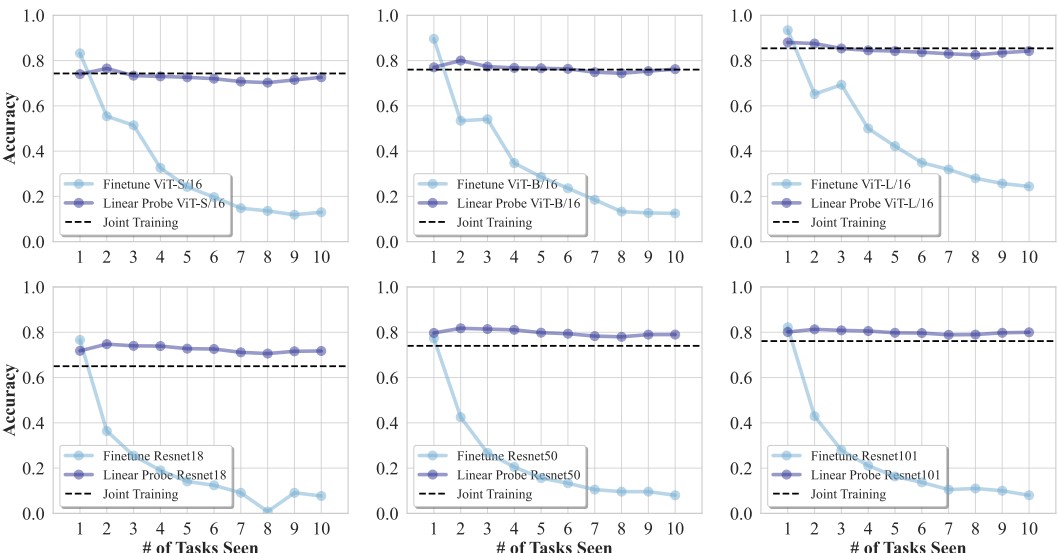

Figure 3: Performance of linear probing with various model architecture and size.

close approximation to probability density, energy score based score functions (LeCun et al., 2006; Wang et al., 2022a) have gained significant prominence. Relating to continual learning, when models train on data of the most current task, data from the previous tasks can be seen as out-of-distribution data, providing us the opportunity to identify these data using OOD techniques.

### 2.3 TEST TIME ADAPTATION

Test Time Adaptation (TTA) focuses on adjusting the model at deployment to accommodate new and possibly shifting data distributions using unlabeled test data Liang et al. (2023); Kundu et al. (2020); Schneider et al. (2020); Sun et al. (2020). In TTA, one of the representative methods is the usage of pseudo-labels (Arazo et al., 2020). For example, in SHOT (Liang et al., 2020), class centroids are obtained by calculating a weighted sum of feature vectors similar to k-means clustering. Then, pseudo-labels are assigned to each test sample according to a nearest centroid classifier. Recent works also investigated a more practical situation of continual test time adaptation (Song et al., 2023), where models are continuously adapted to different target environments. For example, CoTTA (Wang et al., 2022b) introduces weight-averaged and augmentation-averaged pseudo-labels and leverages a teacher model to produce more accurate predictions. However, few works have tried to directly apply TTA approaches to enhance CL methods, and we are among the first to analyze such possibility.

## 3 METHODS

### 3.1 PRELIMINARY NOTATION

In this work, we primarily consider the class-incremental learning problem in the image classification setting, where image data arrives sequentially and models have no access to the task identity during inference. Formally, we consider training the model on a total of $N$ tasks $\mathcal{T} = (T_1, T_2, ..., T_N)$, where each task has a corresponding dataset denoted by $\mathcal{D}(\mathcal{X}^t, \mathcal{Y}^t)$. In class-incremental learning, $\mathcal{Y}^i \cap \mathcal{Y}^j = \emptyset$ and $P_i \neq P_j$ for $i \neq j$, where $P_i$ denotes the probability distribution over samples from task $i$. In other words, each task has distinct classes and a distinct data distribution. A common assumption in class-incremental learning is that the incremental size, i.e., number of classes in each task, remains constant. In this work, we denote this size by $s$.

We define models by

$$M_\theta(\cdot) = h_{\theta_{\text{cls}}}(f_{\theta_{\text{rps}}}(\cdot))$$ (1)

with parameters $\theta = \{\theta_{\text{rps}}, \theta_{\text{cls}}\}$. $f_{\theta_{\text{rps}}}(\cdot)$ represents the feature extractor parameterized by $\theta_{\text{rps}}$, and $h_{\theta_{\text{cls}}}(\cdot)$ denotes the classification layer parameterized by $\theta_{\text{cls}}$.

## 3.2 UNDERSTANDING CATASTROPHIC FORGETTING IN PRETRAINED MODELS

Recent developments in large pretrained models have prompted the continual learning community to also shift towards using these pretrained models, leading to a plethora of well-established works. However, whether the learning dynamics change in this context is not well understood. In this work, we find that the amount of information lost when adapting to a new task is surprisingly minimal. This contrasts with the prevailing expectation in current literature that forgetting would be primarily attributed to changes in the backbone of the model, especially since most current works on pretrained models focus heavily on representation layers. To justify this statement, we leverage models that are fully trained on the latest task and conduct linear probing on data from each preceding task. Here, different from Zhang et al. (2023), we do not pose any constrains on the learning rate, allowing the model to fully adapt to the incoming task.

Specifically, we train models on the Imagenet-R dataset with an incremental step of 20, meaning that there is a total of 10 tasks. After finished training on each new task $T_n$, we perform linear probing on the learned model using data from earlier tasks $T_i$ ($i < n$). We experiment with different architectures and sizes, including ViT-S/16, ViT-B/16, ViT-L/16, and ResNet-18, ResNet-50, ResNet-101. No memory is used throughout the process. The results are shown in Figure 3. We see here that regardless of the specific model architecture or size, the performance of linear probing consistently achieves high performance comparable to joint training. Since linear probing keeps the representation layer fixed, we conclude that the representation layers of pretrained models are adept at retaining knowledge as new tasks are introduced, and they clearly are not the primary source of catastrophic forgetting. Thus, we argue that the key issue for the large performance gap between finetuning and linear probing lies in the bias introduced by the classifier, leading us to the proposed approach ARC.

## 3.3 OUT-OF-TASK DETECTION

In the following, we will build up a method for de-biasing a model's predictions towards the current task. As briefly discussed in the introduction, test samples arrive with no task identity, so the first step towards de-biasing the model's predictions will require accurately distinguishing between data from previous tasks and data from the current task. To do so, we will use the model's own predictions to distinguish cases where a test sample comes from a previous task or the current task.

**Case 1.** For data classified into the classes of past tasks, if the model, after training on the current task, still produces high-confidence predictions, it is highly likely that this data indeed belongs to past tasks. Based on this, we propose the following assumption:

**Assumption 1.** *At task $T_t$, for a test image $\boldsymbol{x}$ with output logits $\boldsymbol{z} \in \mathbb{R}^{s \cdot t}$, let $\beta$ be a pre-defined threshold value. If:*

$$\arg\max_i \frac{\exp \boldsymbol{z}_i}{\sum_{j=1}^{s \cdot t} \exp \boldsymbol{z}_j} \le s \cdot (t-1)$$

*and*

$$c = \max_{1 \le i \le s \cdot t} \frac{\exp \boldsymbol{z}_i}{\sum_{j=1}^{s \cdot t} \exp \boldsymbol{z}_j} \ge \beta,$$

*then we assume $\boldsymbol{x}$ is a sample from the past task that is correctly classified.*

**Case 2.** For data classified into classes of the current task, we need to take more care when considering the model's confidence level. Due to the presence of classifier bias, a low-confidence prediction likely indicates that the test sample comes from a previous task and is misclassified into the current task. However, there is also the likelihood of it being a particularly difficult sample from the current task. To differentiate between the two situations, we introduce a new metric $w$ specifically designed for continual learning, and propose the following assumption:

**Assumption 2.** *At task $T_t$, for a test image $\boldsymbol{x}$ with output logits $\boldsymbol{z} \in \mathbb{R}^{s \cdot t}$, let $\gamma$ be a pre-defined threshold value. If:*

$$\arg\max_i \frac{\exp \boldsymbol{z}_i}{\sum_{j=1}^{s \cdot t} \exp \boldsymbol{z}_j} > s \cdot (t-1)$$

*and*

$$w = \frac{c}{\hat{c}} \le \gamma, \quad where \quad \hat{c} = \max_{1 \le i \le s \cdot (t-1)} \frac{\exp \boldsymbol{z}_i}{\sum_{j=1}^{s \cdot (t-1)} \exp \boldsymbol{z}_j},$$

*then we assume $\boldsymbol{x}$ is a sample from previous tasks misclassified to the current task.*

---

**Algorithm 1** Adaptive Retention & Correction (ARC)

---

**Inputs:** Current task identity $t$; Incremental step size $s$; Testing image $x$; Initially predicted class $\hat{y}$; Adaptive Retention threshold $\beta$; Adaptive Correction threshold $\gamma$; Task-based Softmax Score temperature $T$;network $M_\theta(\cdot) = h_{\theta_{cls}}(f_{\theta_{rps}}(\cdot))$ with parameters $\theta = \{\theta_{rps}, \theta_{cls}\}$;

1: **if** $\hat{y} < s \cdot (t-1)$ **then**
2:     Obtain confidence $c$ of Assumption 1.
3:     **if** $c > \beta$ **then**
4:         Update $\theta_{cls}$ with Equation (3).
5:         Recalculate $\hat{y}$ using the updated classifier.
6:     **end if**
7: **else**
8:     Obtain $w$ of Assumption 2.
9:     **if** $w < \gamma$ **then**
10:        Calculate TSS $\{\mathcal{S}_1, \mathcal{S}_2, \ldots, \mathcal{S}_t\}$ according to Definition 1.
11:        Modify the output according to task $T_i, i = \arg\max_i \mathcal{S}_i$.
12:     **end if**
13: **end if**

---

Here, $\hat{c}$ represents the highest predicted class probability when masking out all the logits corresponding to classes of the current task. The intuition behind this design is that by isolating the logits specific to past tasks from those associated with the current task, we can get a sense of how confident the model is when predicting $x$ as a class from past tasks verses as a class from the current task. Consequently, if the ratio of $c$ and $\hat{c}$ is low, this suggests the model's confidence that $x$ is from the current task is significantly lower than the model's confidence that $x$ is from a past task, and we can assume with high probability that $x$ is indeed a misclassified sample from a past task.

### 3.4 ADAPTIVE RETENTION & CORRECTION

Based on OTD, we design two independent approaches, Adaptive Retention and Adaptive Correction, that leverage the distinct characteristics of the data following Assumption 1 and Assumption 2 to mitigate classifier bias. Together, they form the proposed Adaptive Retention & Correction approach. An overview of ARC is shown in Algorithm 1.

### 3.4.1 ADAPTIVE RETENTION

We first introduce Adaptive Retention using data following Assumption 1. Traditional methods for reducing classifier bias mainly rely on training with data from the past task and are not applicable in memory-free situations. However, by using OTD, we can still leverage data from past tasks, but this time adaptively during the testing phase. To do so, we utilize the predicted labels as supervisory signals and train the classifier with the backbone network frozen using a cross-entropy loss. In this way, the tendency within the classifier towards classes of the current task can be effectively mitigated by re-balancing weights towards previous classes. Furthermore, since predicted labels inevitably introduce errors and uncertainty, we employ an additional entropy minimization loss. This approach enhances the model's confidence in its predictions, thereby offsetting the adverse effects of noise and ensuring more reliable outputs during testing. Finally, to align more closely with practical applications where test samples arrive in an online manner, we choose to train the classifier with only one gradient update on each sample.

Formally, for a test sample $x$ with predicted label $\hat{y}_c$ and following Assumption 1, we obtain the cross-entropy and entropy minimization loss:

$$\mathcal{L}_{CE} = -\hat{y}_c \log p_c \quad \text{and} \quad \mathcal{L}_{EM} = -\sum_i p_i \log p_i, \tag{2}$$

where $p_c$ is the predicted probability for class $\hat{y}_c$, and train the classifier using the combined objective:

$$\mathcal{L}_{CE} + \mathcal{L}_{EM}. \tag{3}$$

### 3.4.2 ADAPTIVE CORRECTION

We now develop a method to correct the model's predictions for test samples following Assumption 2. A naive approach would be to mask out the probabilities corresponding to the current task classes and reassign the sample to the class with the highest probability among the remaining ones. However, such a method defaults to treating all past tasks equally, which we find suboptimal. In reality, in addition to the tendency towards the current task, there is also classification bias among all past tasks, meaning that the classifier would again favor more recent past tasks than earlier ones. To illustrate this phenomenon, again, we refer to the example shown in Figure 2. As is clearly demonstrated, among all the past tasks, the model tends to misclassify images from task $T_1$ into recent past tasks, i.e., task $T_6$, $T_8$ and $T_9$.

Nonetheless, the aforementioned issue can be addressed through a straightforward extension of the naive approach. We first present the following definition:

**Definition 1.** *At task $T_t$, for a test image $x$ with the model output logits $z \in \mathbb{R}^{s \cdot t}$ and a scaling temperature $\mathcal{T}$, we define **Task-based Softmax Score (TSS)** for each task $T_i, i \in \{1, ..., t\}$ as*

$$\mathcal{S}_i = \max_{s \cdot (i-1) \le k \le s \cdot i} \frac{\exp z_k / \mathcal{T}^{(t-i)}}{\sum_{j=1}^{s \cdot i} \exp z_j / \mathcal{T}^{(t-i)}}$$

with the purpose of measuring our confidence that the test sample $x$ originates from task $T_i$. The rationale behind this score is similar to Assumption 2 in that by isolating the logits specific to each task $i$ from those associated with subsequent future tasks, we can minimize classification bias both between past and current tasks and among past tasks themselves, ensuring that the scores obtained can more accurately reflect the true probability for the sample belonging to each task. We also scale the logits using a temperature $\mathcal{T} > 1$ to accommodate the imbalanced denominator for different tasks. Finally, we adjust the prediction accordingly.

## 4 EXPERIMENTS

### 4.1 EXPERIMENTAL SETUP

**Datasets.** We conduct our main experiments on two popular benchmark datasets for continual learning: Split CIFAR-100 and Split Imagenet-R. CIFAR-100 is a relatively simple dataset comprising 100 classes. Imagenet-R, on the other hand, includes 200 classes with data sourced from various styles such as cartoons, graffiti, and origami. The presence of both semantic and covariate shifts makes it one of the most challenging datasets for continual learning. We also conduct experiments on another challenging benchmark, 5-dataset, where unlike the first two, it is a composite of five distinct image classification datasets: CIFAR-10, MNIST, FashionMNIST, SVHN, and notMNIST. The diversity in data distribution across these datasets offers us a more comprehensive evaluation of our method.

**Dataset Split.** We use 'Inc-$n$' to denote the data split setting, where $n$ denotes the number of classes for each incremental stage. For example, Split CIFAR-100 Inc5 means that there are 5 classes for each training step, resulting in a total of 20 tasks. For a fair comparison, we split the classes following the order of Sun et al. (2023).

**Evaluation metrics.** In this work, we mainly use the conventional metrics Average Accuracy and Forgetting. Here, Average Accuracy reflects the overall performance of the model on all tasks after finished sequential training, and Forgetting measures the degree of performance decline on previously learned tasks. Formally, let $R_{t,i}$ be the classification accuracy of task $T_i$ after training on task $T_t$, then Average Accuracy $\mathcal{A}_B$ for task $T_T$ is defined as

$$\mathcal{A}_B = \frac{1}{T} \sum_{i=1}^{T} R_{T,i}, \tag{4}$$

and Forgetting is defined as

$$\mathcal{F} = \frac{1}{T-1} \sum_{i=1}^{T-1} (R_{i,i} - R_{T,i}). \tag{5}$$

| Method | Memory-Free | Split CIFAR-100 Inc5 $\mathcal{A}_\mathcal{B}$ | $\mathcal{F}$ | Split CIFAR-100 Inc10 $\mathcal{A}_\mathcal{B}$ | $\mathcal{F}$ | Split Imagenet-R Inc10 $\mathcal{A}_\mathcal{B}$ | $\mathcal{F}$ | Split Imagenet-R Inc20 $\mathcal{A}_\mathcal{B}$ | $\mathcal{F}$ |
|---|---|---|---|---|---|---|---|---|---|
| Finetune | | 64.6 | 27.2 | 68.4 | 29.8 | 48.8 | 41.3 | 61.5 | 28.6 |
| + ARC | | **66.0** (+1.4) | **24.7** (-2.5) | **71.2** (+2.8) | **10.8** (-19.0) | **50.2** (+1.4) | **35.3** (-6.0) | **63.9** (+2.4) | **15.2** (-13.4) |
| iCarL | | 74.9 | 24.6 | 78.7 | 21.0 | 56.5 | 34.3 | 61.0 | 29.0 |
| + ARC | ✗ | **81.7** (+6.8) | **9.6** (-15.0) | **81.8** (+3.1) | **5.6** (-15.4) | **62.3** (+5.8) | **21.2** (-13.1) | **67.6** (+6.6) | **14.4** (-14.6) |
| Der | | 70.7 | 27.7 | 79.6 | 18.9 | 64.3 | 29.6 | 76.7 | 14.7 |
| + ARC | | **74.0** (+3.3) | **22.3** (-5.4) | **84.1** (+4.5) | **10.3** (-8.6) | **66.0** (+1.7) | **25.8** (-3.8) | **78.4** (+1.7) | **6.4** (-8.3) |
| Memo | | 72.3 | 24.7 | 76.3 | 21.5 | 61.9 | 25.8 | 66.2 | 21.1 |
| + ARC | | **74.7** (+2.4) | **16.5** (-8.2) | **80.0** (+3.7) | **10.3** (-11.2) | **63.1** (+1.2) | **17.7** (-8.1) | **68.2** (+2.0) | **11.9** (-9.2) |
| L2P | | 78.1 | 10.0 | 83.2 | 8.8 | 69.9 | 5.4 | 72.6 | 4.2 |
| + ARC | | **82.8** (+4.7) | **8.0** (-2.0) | **86.2** (+3.0) | **5.7** (-3.1) | **73.3** (+3.4) | **4.7** (-0.7) | **75.1** (+2.5) | **3.5** (-0.7) |
| DualPrompt | | 77.1 | 8.1 | 82.0 | 7.5 | 66.0 | 5.9 | 69.1 | 4.7 |
| + ARC | | **83.4** (+6.3) | **4.4** (-3.6) | **85.0** (+3.0) | **3.8** (-3.7) | **68.7** (+2.7) | **4.2** (-1.7) | **70.5** (+1.4) | **3.5** (-1.2) |
| CodaPrompt | ✓ | 79.6 | **5.7** | 86.9 | 5.1 | 70.5 | 5.6 | 73.2 | 5.7 |
| + ARC | | **81.4** (+1.8) | 5.8 (+0.1) | **88.0** (+1.1) | **4.1** (-1.0) | **72.6** (+2.1) | **2.9** (-2.7) | **75.4** (+2.2) | **2.0** (-3.7) |
| SLCA | | 89.3 | 7.8 | 91.1 | 6.2 | 75.1 | 12.8 | 78.7 | 9.1 |
| + ARC | | **90.4** (+1.1) | **5.9** (-1.9) | **91.8** (+0.7) | **4.3** (-1.9) | **80.4** (+5.3) | **5.3** (-7.5) | **81.3** (+2.7) | **3.5** (-5.6) |

Table 1: Results of incorporating ARC into both classical and state-of-the-art methods on Split CIFAR-100 and Split Imagenet-R. We mainly use the implementation provided by Sun et al. (2023).

**Implementation details.** We use the repository provided by Sun et al. (2023) to test the effectiveness of our method. This repository is a pre-trained model-based continual learning toolbox consisting of both classical and most recent state-of-the-art methods. Specifically, we test ARC by incorporating it with Finetune, iCarL (Rebuffi et al., 2017), Memo (Zhou et al., 2022), Der (Yan et al., 2021), L2P (Wang et al., 2022d), DualPrompt (Wang et al., 2022c), CodaPrompt (Smith et al., 2023), and SLCA (Zhang et al., 2023). Finetune, iCarL, Memo, and Der are equipped with a memory buffer of 2000 and 4000 for Split CIFAR-100 and Split Imagenet-R, respectively, whereas L2P, DualPrompt, CodaPrompt and SLCA are memory-free. For all tested methods, we utilize a ViT-b-16 backbone pretrained on Imagenet-21K. We set the training configuration to be the same as provided.

## 4.2 COMBINING ARC WITH STATE-OF-THE-ART CONTINUAL LEARNING METHODS

**Results on benchmark with shared distribution.** We report results on Split CIFAR-100 and Split Imagenet-R in Table 1. These two datasets, in their entirety, are drawn from the same underlying distribution. We test our methods on both short and long sequence where there is a total of 10 and 20 tasks, respectively. We see here that integrating ARC can significantly improve the Average Accuracy and reduce the Forgetting of state-of-the-art methods. Specifically, for short sequence, all methods experience an average $\mathcal{A}_\mathcal{B}$ boost of 2.7% on both Split CIFAR-100 and Split Imagenet-R. Similarly, the drop for $\mathcal{F}$ reaches an average of 8.0% and 7.0%. On the other hand, for long sequence, all methods experience an average $\mathcal{A}_\mathcal{B}$ boost of 3.5% and 3.0%, and an average $\mathcal{F}$ drop of 4.8% and 5.5%, respectively. We note that the overall improvements is more significant on long sequence, indicating that ARC is more compatible with scenarios where the task number is large.

| Method | 5-dataset Inc10 $\mathcal{A}_\mathcal{B}$ | $\mathcal{F}$ |
|---|---|---|
| iCarL | 89.8 | 5.0 |
| + ARC | **91.9** (+2.1) | **2.9** (-2.1) |
| Der | 95.9 | 1.1 |
| + ARC | **96.8** (+0.9) | **0.1** (-1.0) |
| Memo | 91.9 | 3.8 |
| + ARC | **93.0** (+2.1) | **0.6** (-3.2) |
| L2P | 82.5 | 4.9 |
| + ARC | **87.8** (+5.3) | **2.4** (-2.5) |

Table 2: ARC on 5-dataset.

| Method | Split CIFAR-100 Inc10 $\mathcal{A}_\mathcal{B}$ | $\mathcal{F}$ | Split Imagene-R Inc20 $\mathcal{A}_\mathcal{B}$ | $\mathcal{F}$ |
|---|---|---|---|---|
| iCarL | 78.7 | 21.0 | 61.0 | 29.0 |
| + TENT | 80.1 (+1.4) | 12.4 (-8.6) | 61.6 (+0.6) | 21.9 (-7.1) |
| + ARC | **81.8** (+3.1) | **5.6** (-15.4) | **67.6** (+6.6) | **14.4** (-14.6) |
| Der | 79.6 | 18.9 | 76.7 | 14.7 |
| + TENT | 81.8 (+2.2) | 16.0 (-2.9) | 77.1 (+0.4) | 11.2 (-3.5) |
| + ARC | **84.1** (+4.5) | **10.3** (-8.6) | **78.4** (+1.7) | **6.4** (-8.3) |
| Memo | 76.3 | 21.5 | 66.2 | 21.1 |
| + TENT | 77.1 (+0.8) | 18.9 (-2.6) | 65.8 (-0.4) | 21.3 (+0.2) |
| + ARC | **80.0** (+3.7) | **10.3** (-11.2) | **68.2** (+2.0) | **11.9** (-9.2) |
| L2P | 83.2 | 8.8 | 72.6 | 4.2 |
| + TENT | 86.0 (+2.8) | 4.9 (-3.9) | 73.2 (+0.6) | **2.7** (-1.5) |
| + ARC | **86.2** (+3.0) | 5.7 (-3.1) | **75.1** (+2.5) | 3.5 (-0.7) |

Table 3: Performance comparison with TENT.

**Results on benchmark with more diverse distribution.** We report results on 5-dataset in Table 2. Unlike the other two benchmarks, the data in 5-dataset exhibit a much more diverse distribution. Still,

incorporating ARC into existing methods leads to notable improvements, where $\mathcal{A}_{\mathcal{B}}$ increases by an average of 2.6%, and $\mathcal{F}$ decreases by an average of 2.2%. This highlights the robustness and efficacy of ARC across different benchmark scenarios, making it valuable for practical usage.

**Comparison with TTA method.** Since Test Time Adaptation (TTA) is a special type of model correction, we also report performance comparison with a representative TTA method, TENT (Wang et al., 2020) in Table 3. As demonstrated, while TENT exhibits some performance improvements, the magnitude is consistently lower compared to ARC. Specifically, on Split CIFAR100 Inc10, TENT's average performance gains are 1.8% lower than those of ARC, and on Split ImageNet-R Inc20, the gap increases to 2.9%. We hypothesize that there are two primary reasons for this. First, TENT, and similar TTA methods, adapt to all incoming test data, whereas our approach selectively adapts only to past task data. Second, TENT's adaptation is comparable to Adaptive Retention but lacks the ability to identify and correct potentially misclassified data from previous tasks, i.e., Adaptive Correction, further limiting its effectiveness.

## 4.3 IN-DEPTH ANALYSIS OF ARC AND ITS PROPERTIES

**Detailed contribution.** We further present the specific contributions of Adaptive Retention and Adaptive Correction individually in Table 4. The average increase using Adaptive Retention is 1.8% on Split CIFAR-100 and Split Imagenet-R, and the average increase using Adaptive Correction is 1.1% on Split CIFAR-100 and 1.2% on Split Imagenet-R. Notably, the performance boost when applying Adaptive Correction to prompt-based methods is low. We find that the small size of this boost is due to the small number of test samples for these prompt based methods that satisfy Assumption 2. For example, on Split Imagenet-R, only 1.9% of test samples satisfy Assumption 2 for L2P, while for iCaRL, 7.4% of test samples satisfy this assumption.

| Method | Split CIFAR-100 | | Split Imagenet-R | |
|---|---|---|---|---|
| | Ada. Self-Ret. | Ada. Self-Cor. | Ada. Ret. | Ada. Cor. |
| Finetune | +2.4 | +0.4 | +0.2 | +2.3 |
| iCaRL | +2.9 | +0.2 | +4.6 | +1.9 |
| Der | +0.6 | +3.9 | +0.2 | +1.5 |
| Memo | +1.4 | +2.3 | +1.4 | +0.6 |
| L2P | +2.7 | +0.3 | +2.0 | +0.2 |
| DualPrompt | +2.6 | +0.4 | +1.4 | +0.0 |
| CODA-Prompt | +1.0 | +0.1 | +1.9 | +0.3 |
| SLCA | +0.4 | +0.2 | +2.0 | +0.7 |

| Method | Split Imagenet-R | |
|---|---|---|
| | Assump. 1 Accuracy | Assump. 2 Accuracy |
| Finetune | 83.2% | 70.7% |
| iCaRL | 92.4% | 86.5% |
| Der | 91.8% | 83.2% |
| Memo | 85.6% | 70.6% |
| L2P | 87.7% | 60.0% |
| DualPrompt | 81.6% | 68.6% |
| CODA-Prompt | 92.8% | 62.4% |
| SLCA | 92.1% | 73.3% |

Table 4: Specific contribution of Adaptive Retention and Adaptive Correction.

Table 5: Empirical validation for Assumptions 1 and 2 on Split Imagenet-R.

**Empirical validation of assumptions 1 and 2.** To validate Assumption 1 and Assumption 2, we present the empirical results on Split Imagenet-R in Table 5, where the reported results indicate how well our assumptions hold. The proportion of data following Assumption 1 that are indeed samples from previous tasks correctly classified by the model is 88.4% averaged on all methods. Likewise, the proportion of data following Assumption 2 that are indeed samples from previous tasks misclassified to the current task is 71.9% averaged on all methods. These results demonstrate the overall effectiveness of the proposed assumptions.

## 4.4 ABLATION STUDIES

**Adaptive Retention.** We present in Figure 4a the ablation studies on the objective of the training process of Adaptive Retention, where the effect of the cross-entropy loss $\mathcal{L}_{CE}$ and the entropy minimization loss $\mathcal{L}_{EM}$ on the Split CIFAR-100 dataset is tested. As is shown, while for L2P, the use of $\mathcal{L}_{CE}$ results in a slightly worse performance, in general, both are important components of Adaptive Retention.

**Adaptive Correction.** We further present in Figure 4b the ablation studies on our design of Adaptive Correction. We first validate the necessity of the scaling temperature value for TSS. We see that when it is excluded, Finetune, iCaRL, Der and Memo would experience an average performance drop of 0.6%. We also test our design of $w$ for Assumption 2. Specifically, we examine whether our concern over the two possible outcomes of low-confidence prediction is necessary. To do so, we compared against letting $w = c \leq \gamma$, i.e., making $w$ represent the raw confidence for the predicted class. As is

| Method | $\mathcal{L}_{CE}$ | $\mathcal{L}_{EM}$ | Average Accuracy |
|--------|--------|--------|------------------|
| L2P | ✓ | ✗ | 79.6 |
| | ✗ | ✓ | **86.6** |
| | ✓ | ✓ | 86.2 |
| DualPrompt | ✓ | ✗ | 83.4 |
| | ✗ | ✓ | 84.5 |
| | ✓ | ✓ | **85.0** |
| CodaPrompt | ✓ | ✗ | 87.1 |
| | ✗ | ✓ | 86.6 |
| | ✓ | ✓ | **88.0** |
| SLCA | ✓ | ✗ | 91.4 |
| | ✗ | ✓ | 91.7 |
| | ✓ | ✓ | **91.8** |

(a) Ablation for Adaptive Retention.

| Method | $\mathcal{T}$ | $w$ | Average Accuracy |
|--------|--------|--------|------------------|
| Finetune | ✓ | ✗ | 70.5 |
| | ✗ | ✓ | 70.5 |
| | ✓ | ✓ | **71.2** |
| iCarL | ✓ | ✗ | 81.3 |
| | ✗ | ✓ | 81.6 |
| | ✓ | ✓ | **81.8** |
| Der | ✓ | ✗ | 83.2 |
| | ✗ | ✓ | 83.5 |
| | ✓ | ✓ | **84.1** |
| Memo | ✓ | ✗ | 79.1 |
| | ✗ | ✓ | 79.7 |
| | ✓ | ✓ | **80.0** |

(b) Ablation for Adaptive Correction.

Figure 4: Ablation studies for specific components of Adaptive Retention and Adaptive Correction on Split CIFAR-100.

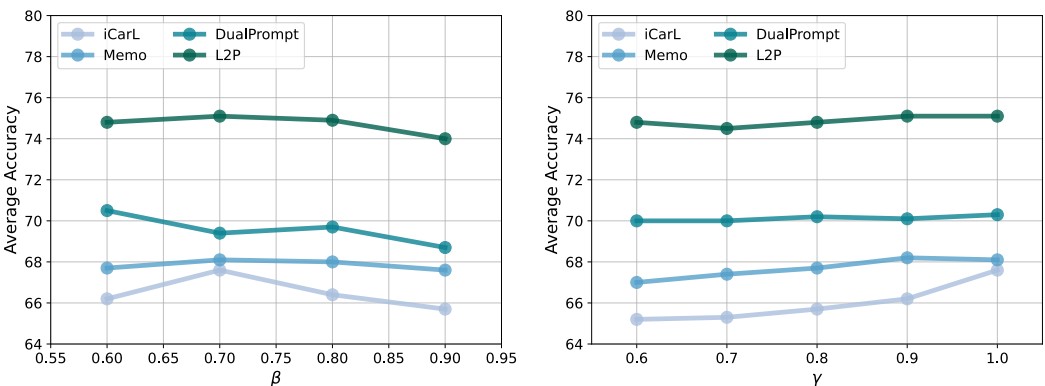

Figure 5: Ablation study on threshold $\beta$ and $\gamma$ of Assumption 1 and Assumption 2.

demonstrated, such change would result in an average performance degradation of 0.5%, indicating the success of our specific design of $w$.

**Hyperparameter.** Figure 5 depicts the ablation study for hyperparameter $\beta$ and $\gamma$ of Assumption 1 and Assumption 2, where we experiment on 4 of the tested methods (iCarL, DualPrompt, Memo and L2P). We test $\beta$ ranging from 0.6 to 0.9, and $\gamma$ from 0.6 to 1.0. As is clearly shown, the performance shows relatively small fluctuation when using different threshold values, again indicating the robustness of ARC.

## 5 CONCLUSION

In this study, we addressed the critical issue of classifier bias in continual learning, especially in the current paradigm of leveraging pretrained models under memory-free settings, during the testing phase. We first built an Out-of-Task detection method, where we identified two special cases of testing data: 1) data from previous tasks that is correctly classified and 2) data from previous tasks that is misclassified to the current task. Consequently, we proposed two independent approaches, Adaptive Retention and Adaptive Correction, specifically tailored for each case. Together they form our approach Adaptive Retention & Correction (ARC). Adaptive Retention focuses on dynamically tuning the classifier layer and Adaptive Correction implements a training-free bias correction mechanism. By leveraging the ability to access past task data during inference, ARC effectively mitigates classifier bias without requiring any modifications to the training procedure. Extensive experiments show that integrating ARC to existing methods can significantly boost their overall performance, demonstrating the effectiveness and robustness of our method.

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
