# A   ADDITIONAL IMPLEMENTATION DETAIL.

In this section, we provide further details to ensure the reproducibility of our work. Specifically, for Adaptive Retention, given a batch of test samples, we compute the loss using Equation (3) and perform a single gradient update based on this loss. For non-prompt-based methods, we utilize the SGD optimizer, while for prompt-based methods, we employ the Adam optimizer.

Conceptually, this approach is equivalent to training on the test data for a single epoch, where each test sample is used exactly once. This design of performing only one gradient update aligns with our objective of simulating realistic scenarios. In standard testing, models process a batch of test samples in a single forward pass to generate predictions. In ARC, we extend this principle by limiting the process to one forward pass and one gradient update, thereby preserving both the efficiency and practicality of the testing procedure.

For details on hyperparameter settings, we refer readers to our codebase.

# B   ADDITIONAL COMPARISON WITH TTA METHOD.

We also attempt to compare with the CoTTA Wang et al. (2022b) method. However, our findings reveal that CoTTA is not compatible with prompt-based methods. Directly applying CoTTA to these methods leads to a decrease in performance, as shown in Table 6. For non-prompt-based methods, while CoTTA does provide some improvements, the gains are less significant than those achieved by ARC. This is evident in the comparison of Table 7.

| Method | Split Imagenet-R Inc 20 $\mathcal{A}_B$ |
|---|---|
| L2P | 72.6 |
| L2P + CoTTA | 72.2 |
| DualPrompt | 69.1 |
| DualPrompt + CoTTA | 67.7 |

Table 6: CoTTA on prompt based methods.

| Method | Split Imagenet-R Inc 20 $\mathcal{A}_B$ |
|---|---|
| iCarL + CoTTA | 64.7 |
| iCarL + ARC | 67.6 |
| Memo + CoTTA | 67.6 |
| Memo + ARC | 68.2 |

Table 7: CoTTA on non-prompt based methods.

# C   TIME COMPLEXITY.

We also perform time complexity analysis in Table 8. As shown, there is a certain amount of additional inference time required for ARC. This is expected, as ARC is specifically designed to address classification bias during inference, which naturally incurs some additional computational cost. Furthermore, it is worth noting that research in other areas, such as large language models (LLMs), are increasingly exploring methods that enhance performance by utilizing additional inference-time computation. There is a growing consensus that additional inference time is a reasonable trade-off when it leads to significant improvements in model performance. Similarly, we believe the computational cost of our method is well-justified by the substantial performance gains it delivers.

| Method | Split Imagenet-R Inc 20 | |
|---|---|---|
| | Additional Time | Performance Gain |
| iCarL | 11% | 6.6 |
| Memo | 8% | 2.0 |
| L2P | 37% | 2.5 |
| DualPrompt | 32% | 1.4 |

Table 8: Additional time requirement for ARC.

| Method | Split Imagenet-R Inc 20 $\mathcal{A}_B$ |
|---|---|
| iCarL + ARC | 64.7 |
| iCarL + ARC-Last | 67.6 |
| Memo + ARC | 67.6 |
| Memo + ARC-Last | 68.2 |
| L2P + ARC | 64.7 |
| L2P + ARC-Last | 67.6 |
| DualPrompt + ARC | 67.6 |
| DualPrompt + ARC-Last | 68.2 |

Table 9: ARC applied on last task only.

## D ARC ONLY ON FINAL TASK.

Currently, we apply ARC after completing training on each task. However, it is also valuable to explore how ARC performs when applied only after the final training task. In Table 9, we denote this approach as ARC-Last. As shown, ARC continues to yield significant improvements even when applied exclusively after the final task, further demonstrating the effectiveness of our approach.