# OpenReview forum: "Adaptive Retention & Correction: Test-Time Training for Continual Learning"
_ICLR.cc/2025/Conference — ICLR 2025 Poster_

### Official Review · Reviewer_hK7E · 2024-10-31

**Soundness:** 3
**Presentation:** 2
**Contribution:** 3
**Rating:** 6
**Confidence:** 4

**Summary:**

This paper proposes to reduce the task-recency bias present in the last layer when training in a Continual Learning environment. This well-known problem remains under-explored in the case of memory-free methods and presents additional challenges. In particular, the authors propose to leverage test data in order to adapt the weights of the last layer by differencing old and newer data using prediction confidence. The authors also propose an alternative to the softmax to rebalance the model’s logits, which as often bias toward later tasks. Eventually, this paper shows significant improvement by combining their approach with current state-of-the-art methods.

**Strengths:**

- The idea of focusing on the debiasing the last layer for later prompt-based memory-free methods is interesting and under-explored
- The intuition behind focusing on test-time adaptation by taking into context the specifics behaviour of continually trained model is good and intuitive
- The method performances are appealing and it seems quite robust to hyper-parameter, which is crucial in Continual Learning where hyper-parameter search is unrealistic
- Most sections are well-written and easy to follow

**Weaknesses:**

### Major weaknesses
- There is a lack of detail on the overall training procedure. How many epochs are used for training? Is this correction and retention procedure applied after each task? In that case, this would correspond to storing unlabeled memory data.
- I don’t think the formulae in definition 1 is well defined. The inequality under the max does not make much sense as $s(i-1) \leq i \leq s.i$ is impossible in many cases. Also, taking $T>1$ implies that the output of the softmax will be lower (flattened) for earlier tasks (low $i$ values), but I believe the objective is the opposite; the authors want to obtain higher softmax values for earlier tasks. I also do not really understand the usage of a maximum operator here. Overall, this definition is very confusing.
- There is a lack of detail on the overall training procedure. How many epochs are used for training? Is this correction and retention procedure applied after each task? In that case, this would correspond to storing unlabeled memory data.
- This method is connected to Test-Time Adaptation and some discussion in the related work would be appreciated. While TTA methods are rightfully compared in section 4.2, such discussion could be included more thoroughly in the related work.
- The impact of the temperature is not presented in the paper
- The notation for the temperature and the number of tasks is the same, $T$, which is confusing
- in Figure 1, is the independent classifier trained for doing classification of one task only? Then is the problem a TIL problem? If this is the case, the performances are comparing a TIL problem to a CIL problem, which is unfair since the CIL problem is much harder than the TIL problem.
- l. 202 the distribution are defined as different between tasks but in 4.2, the incremental dataset are defined as IID. I believe this is a mistake, can you elaborate?
### Minor weaknesses
- The experiments of figure 3 is quite unclear. It depends on the optimization strategy and the number of epochs for instance. Also, do you use the same fixed learning rate for joint training?
- I believe a confusion matrix would be more clearer than Figure 1 at least, maybe even Figure 1 and 2
- How much data do you use for linear probe?
- I believe the difference in performances in figure 3 between finetune and probe makes sense and justifies the claim that the representation knowledge is still transferable across task.  However, I do not see why the joint training performances are lower in most cases. Could you elaborate?
- the code is not shared
### typos
- The title is spelled wrong.
If the authors can clarify my concerns, I would happily increase my score.

**Questions:**

See weaknesses.

---

> ### Author Response · Authors · 2024-11-20
> **Response from authors**
>
> We thank reviewer hK7E for the thoughtful and constructive comments. It is evident that the reviewer spent considerable time carefully reading our work and provided valuable insights and suggestions. We really appreciate the detailed feedback and the recognition of the strengths in our approach. Below, we address the specific points raised by the reviewer.
>
> >*There is a lack of detail on the overall training procedure. How many epochs are used for training?*
>
> We apologize for any confusion. In fact, ARC is not involved in the training process. It is only applied during inference, and thus, details such as the number of training epochs are not applicable. During inference, the classifier is adaptively rebalanced for incoming test samples. This process involves only one gradient update per sample, performed in an online manner. In other words, we only use each sample once for classifier tuning.
>
> >*Is this correction and retention procedure applied after each task? In that case, this would correspond to storing unlabeled memory data.*
>
> Yes, this correction and retention procedure is applied after each task. However, we do not store any unlabeled memory data. Instead, our approach relies solely on on-the-fly test samples during inference.
>
> >*Confusion on the formula of Definition 1*
>
> Thank you for pointing out the confusion on $i$ in the max operator. We apologize for the typo in the original definition. The corrected definition is as follows:
>
> $S_i = \max_{s \cdot (i-1) \leq k < s \cdot i} \frac{ \exp{z_k / T^{(t - i)}}}{\sum_{j=1}^{s \cdot i} \exp{z_j / T^{(t - i)}}}$
>
> To clarify this definition, we can compare it to the standard softmax calculation. In the standard softmax, the numerator involves the logits for all classes. However, in our case, we focus specifically on the highest logit for task $i$, which is why the max operator is used in the numerator. Similarly, for the denominator, instead of summing over logits for all classes, we restrict the sum to only the classes revealed to the model up to task $i$, which is also the reason why we have a temperature value $T > 1$ ("accommodate the imbalanced denominator for different tasks" as stated in L344). In this way we can minimize classification bias both between past and current tasks and among past tasks themselves.
>
> >*This method is connected to Test-Time Adaptation and some discussion in the related work would be appreciated. While TTA methods are rightfully compared in section 4.2, such discussion could be included more thoroughly in the related work.*
>
> Thank you for the suggestion! While we briefly discussed TTA methods in Section 2.3, we agree that this section can be expanded to provide a more thorough discussion. In the revised version, we’ll make sure to include a more detailed discussion to better connect our work with TTA methods.
>
> >*The impact of the temperature is not presented in the paper*
>
> Thank you for pointing this out. In the table below we present ablation results on the impact of the temperature value. As shown, incorporating the temperature parameter improves performance consistently:
>
> ||Temperature |Avg Acc on Imagenet-R Inc20|
> |----:|:---:|:---:|
> |iCarL |&check;  |67.6  |
> |iCarL |&cross;  |67.4  |
> |Memo  |&check;    |68.2  |
> |Memo  |&cross;    |67.8  |
> |L2P  |&check; |75.1 |
> |L2P  |&cross;  |74.6 |
> |DualPrompt |&check; |70.5|
> |DualPrompt |&cross;  |69.9 |
>
> >*The notation for the temperature and the number of tasks is the same which is confusing*
>
> Thank you for pointing this out! These are supposed to be different notations. We will revise accordingly.
>
> >*In Figure 1, is the independent classifier trained for doing classification of one task only? Then is the problem a TIL problem? If this is the case, the performances are comparing a TIL problem to a CIL problem, which is unfair since the CIL problem is much harder than the TIL problem.*
>
> Yes, you are correct that the independent classifier in Figure 1 is trained under the TIL setting, while the shared classifier is trained under the CIL problem. In fact, the purpose of Figure 1 is to demonstrate why the CIL problem is significantly harder than the TIL problem, i.e., primarily due to classifier bias. This highlights how the way classifiers are constructed can have a much greater impact on overall continual learning performance compared to the backbone architecture, which is a major motivation of our approach. We included this figure to bring attention to a key point that might not be immediately obvious to readers less familiar with continual learning.

---

> > ### Author Response · Authors · 2024-11-20
> > **Response from authors**
> >
> > >*l. 202 the distribution are defined as different between tasks but in 4.2, the incremental dataset are defined as IID. I believe this is a mistake, can you elaborate?*
> >
> > Thank you for pointing this out! We agree that the use of the word IID is not rigorous and will find a better description. To clarify, when we describe the incremental datasets as IID in Section 4.2, we are referring to the dataset as a whole being drawn from an identical underlying distribution. In contrast, in the 5-dataset setting, the overall dataset exhibits a more significant and complex distribution shift, as it combines data from different datasets with distinct characteristics.
> >
> > For l. 202, we are referring to scenarios where a dataset is split into subsets with disjoint label spaces (e.g., distinct classes). In such cases, the distributions between these subsets differ due to the label-space partitioning.
> >
> > This distinction was our intended meaning, and we will clarify this more explicitly in the revised version.
> >
> > >*The experiments of figure 3 is quite unclear. It depends on the optimization strategy and the number of epochs for instance. Also, do you use the same fixed learning rate for joint training?*
> >
> > For all experiments in Figure 3, we did not use the same optimization strategy, number of epochs, or learning rate. Instead, each method is trained individually to ensure full convergence.
> >
> > >*I believe a confusion matrix would be more clearer than Figure 1 at least, maybe even Figure 1 and 2*
> >
> > Thanks for the suggestion! We will modify accordingly.
> >
> > >*How much data do you use for linear probe?*
> >
> > All of the training data is used.
> >
> > >*I believe the difference in performances in figure 3 between finetune and probe makes sense and justifies the claim that the representation knowledge is still transferable across task. However, I do not see why the joint training performances are lower in most cases. Could you elaborate?*
> >
> > Great point! There are several possibilites:
> > - Since Imagenet-R is a relatively hard dataset, joint training might struggle to balance learning across different data subsets. In contrast, sequential training with linear probing could mitigate interference by allowing the model to specialize on each subset and then use linear probing to combine the learned features effectively.
> > - Joint training could face optimization difficulties due to things like conflicting gradients from subsets of the data, wheres sequential training with linear probing avoids this by breaking down the optimization problem into smaller, more tractable stages.
> > - Linear probing introduces an additional constraint by freezing the feature extractor, which can act as a form of regularization. This separation can lead to a more robust classifier compared to joint end-to-end optimization.
> >
> > Moreover, in Figure 3, it is evident that for ViTs, the performance of joint training and linear probing is relatively similar. However, for ResNets, particularly as the model scale decreases, joint training consistently underperforms compared to linear probing. This observation aligns with the explanations above, as smaller models are more prone to optimization challenges and interference during joint training, which linear probing can potentially help alleviate.
> >
> > >*The code is not shared*
> >
> > We promise to release the code upon acceptance.
> >
> > >*Typo in the title*
> >
> > Thank you so much for pointing this out! We will correct it in the revised version.

---

> ### Author Response · Authors · 2024-11-24
> **Response from authors**
>
> Dear reviewer, we would be grateful if you could confirm whether our response has addressed your concerns. Please do not hesitate to let us know whether there is anything else you would like to see clarified or improved before the end of the rebuttal period.

---

> > ### Comment · Reviewer_hK7E · 2024-11-24
> > **Thank you for your rebuttal**
> >
> > I thank the authors for their thorough response, which clarifies most of my concerns. However, there is still one point which I am not sure to understand and is very important for me to quantify the overall contribution of this paper. I will start with this *critical* point first.
> >
> > > Yes, this correction and retention procedure is applied after each task. However, we do not store any unlabeled memory data. Instead, our approach relies solely on on-the-fly test samples during inference.
> >
> > Let us consider the weights of the classification layer, defined as $\theta_{cls}$ in the paper. Now, these weights are updated after each task, let us say that $\theta_{cls}^t$ are the corresponding classifier weight values for a task $t$. Then I think we can agree that $\theta_{cls}^{t+1}$ are computed using Eq 3, starting from the value of $\theta_{cls}^t$ and using **all the testing data from previous task**. This means that you use test data belonging to previous tasks when updating for the current task, and this is done **at the end of each task during training**. This is very different than training a model once, than adapting once during testing. Here you adapt several times at every test iteration, so, unless my understanding is incorrect, you strategy is equivalent to having access to unlabeled memory data at the end of each task.
> > Now, I still consider your work to be interesting and I do like the approach, however, I believe that it cannot be described as “memory free” in its current stage, unless you apply the procedure **only once at the end of training**.
> > Unfortunately, I believe this to be a critical point and I cannot raise my score. **I think the paper cannot be accepted as long as it is presented as a “memory-free” approach without further development regarding the above issue**.
> >
> > Regarding others remarks:
> > > We apologize for any confusion. In fact, ARC is not involved in the training process. It is only applied during inference, and thus, details such as the number of training epochs are not applicable. During inference, the classifier is adaptively rebalanced for incoming test samples. This process involves only one gradient update per sample, performed in an online manner. In other words, we only use each sample once for classifier tuning.
> >
> > I understand that ARC occurs during testing only, however, I do not see any details regarding the training procedure of the methods combined with ARC. I believe this including such detailed in the appendix would help the reader have a clear understanding of your training procedure.
> >
> > Other issues I mentioned have been addressed by the authors and I sincerely thank them for their time and effort throughout this rebuttal.

---

> ### Author Response · Authors · 2024-11-25
> **Response from authors**
>
> Thank you for your valuable feedback! Regarding the most critical point, we believe there might be a misunderstanding of the continual learning (CL) setting. Specifically, in real-world CL use cases:
>
> 1. A model is initially trained on a dataset and deployed in practice.
> 2. Over time, new requirements emerge or new data becomes available, necessitating updates to the model.
> 3. The updated model is redeployed and continues to be used in practice.
> 4. This process repeats as additional data or requirements arise.
>
> ARC specifically operates during Step 3 and does not require storing any past-task data. To illustrate, consider a scenario where we initially train a model to distinguish between cats and dogs. Later, we may want to add a new class, such as birds, to the model (as our approach focuses on class-incremental learning). After training the model on bird data and redeploying it, the model’s practical application is to classify among cats, dogs, and birds. Naturally, during deployment, the model will encounter samples from previous tasks (cats and dogs), allowing ARC to be utilized without storing any past-task data.
>
> Subsequently, we might add another class, such as turtles. After updating the model to include this new class, it is redeployed to classify among four types of animals. Once again, ARC can be applied in a similar manner during deployment, without requiring any storage of past-task data. This cycle reflects typical deployment scenarios, and we believe this is how ARC can be applied after every task in practice.
>
> Furthermore, there is also a metric in CL that refers to the **averaged average accuracy**:
> $
> \bar{A} = \frac{1}{N}A_B
> $
> , where $A_B$ is defined in equation 4 of our paper. This metric explicitly measures cumulative performance by evaluating the model after each task, further validating our design choice of applying our method after each sequential training phase.
>
> Therefore, regarding your concern that our approach implicitly stores past-task memory data, we respectfully disagree. Our method does not store any data from past tasks. Instead, as part of the natural process of being deployed in practice, the model may encounter past-task samples, enabling our method to function without violating the “memory-free” claim.
>
> Nonetheless, we greatly appreciate your suggestion and have conducted additional experiments under the proposed setup, where our method is applied **only after the final training task**. The results are presented in the table below:
>
> ||Avg Acc on Imagenet-R Inc20|
> |----:|:---:|
> |iCarL   |61.0  |
> |iCarL + ARC (after each task)  |67.6  |
> |iCarL + ARC (only after the final task) |67.1  |
> |Memo  |66.2 |
> |Memo + ARC (after each task) |68.2 |
> |Memo + ARC (only after the final task) |67.1 |
> |L2P  |72.6 |
> |L2P + ARC (after each task) |75.1 |
> |L2P + ARC (only after the final task)|74.3 |
> |DualPrompt   |69.1  |
> |DualPrompt + ARC (after each task) |70.5  |
> |DualPrompt + ARC (only after the final task) |69.9  |
>
> As shown in the results, ARC still provides significant improvements even when applied only after the final training task. We stronly believe that these findings support the effectiveness of our approach.
>
> >*Details regarding the training procedure*
>
> We appreciate your suggestion and will include these details in the appendix to provide readers with a clearer understanding of our training procedure.
>
> In summary, for non-prompt-based methods, we use the SGD optimizer, and for prompt-based methods, we employ the Adam optimizer. During testing, given a batch of test samples, we calculate the loss using Equation 3 from the paper. Based on this loss, we perform a single gradient update. Theoretically, this process is equivalent to training on the test data for just 1 epoch, where each test sample is used exactly once.
>
> We want to emphasize that this design of performing only one gradient update aligns with our goal of simulating realistic scenarios. In standard testing, a model processes a batch of test samples with a single forward pass to produce predictions. In ARC, we adhere to this principle by limiting the process to a single forward pass followed by one gradient update, maintaining the efficiency and practicality of the testing procedure.
>
> We sincerely thank you once again for your thoughtful feedback. If there is anything else that requires clarification, please let us know before the rebuttal period concludes.

---

> > ### Comment · Reviewer_hK7E · 2024-11-26
> > **Thanks you for the information**
> >
> > I deeply thank the authors for taking the time to explain and clarify their work throughout this rebuttal.
> >
> > > ARC specifically operates during Step 3 and does not require storing any past-task data. To illustrate, consider a scenario where we initially train a model to distinguish between cats and dogs. Later, we may want to add a new class, such as birds, to the model (as our approach focuses on class-incremental learning). After training the model on bird data and redeploying it, the model’s practical application is to classify among cats, dogs, and birds. Naturally, during deployment, the model will encounter samples from previous tasks (cats and dogs), allowing ARC to be utilized without storing any past-task data.
> >
> > I see your point and agree with the practicality of such setup. I guess what is debatable is whether you store those “data seen during deployment”. But you could certainly consider that new deployment data of any task come on a regular basis. It would be interesting for future work to consider cases where this test data comes in a separate fashion, e.g., an “incremental test time adaptation”. In any case, I agree with the authors and would like to thank them again for their explanation. Maybe such practical example could be included in the introduction to further improve the quality of the paper.
> >
> > > Nonetheless, we greatly appreciate your suggestion and have conducted additional experiments under the proposed setup, where our method is applied only after the final training task. The results are presented in the table below:
> >
> > To follow up on my previous comment, I really appreciate these experiments which for me remove any doubt regarding the practicality of such method. I advise the authors to include such interesting results in the main draft or the appendix.
> > Overall, I agree with the authors and believe this work to be valuable for the Continual Learning community, the main issue I have with the current version of the paper is the presentation. Improving the presentation as per our discussions would definitely improve the overall quality of the paper. In that sense, I will raise my score to 6.

---

> > > ### Author Response · Authors · 2024-11-26
> > > **Response from authors**
> > >
> > > We sincerely thank you for taking the time to engage with our work so thoroughly and for providing such thoughtful feedback! Your suggestion to include these results in the main draft or appendix is greatly appreciated, and we will incorporate them into the revised version. We are also grateful for your support and for raising your score. Your insights have been invaluable in helping us refine and strengthen our work.

---

### Official Review · Reviewer_EsC3 · 2024-11-02

**Soundness:** 2
**Presentation:** 2
**Contribution:** 2
**Rating:** 6
**Confidence:** 4

**Summary:**

This paper addresses the challenge of reducing classification bias in class incremental learning (CIL) by utilizing test samples. The authors introduce a novel method called Adaptive Retention & Correction (ARC), which dynamically adjusts classifier layers using test samples confidently identified as belonging to previous tasks. Additionally, ARC corrects test samples that are mistakenly classified as part of the current task.

**Strengths:**

- **Originality:** The method of adaptively adjusting and correcting model predictions using test sample confidence is novel.
- **Quality:**  Leveraging confidence scores to detect past task samples and using these pseudo-labeled samples to train the model represents a reasonable strategy for mitigating classification bias in CIL.

**Weaknesses:**

1. While the motivation to correct misclassified past task samples is sound, the proposed Task-based Softmax Score (TSS) may be problematic. For the earlier tasks, as the temperature $T^{(t−i)}$ increases, the probability distribution flattens, which could hinder the model’s ability to recognize samples from past tasks.

2. Although the experiments demonstrate the effectiveness of ARC, the paper lacks comparisons with other test-time adaptation (TTA) benchmarks, such as CoTTA. Additionally, the introduction of extra data for classifier training (referred to as retention) raises concerns about increased training costs that should be addressed.

3. The paper lacks clarity in several areas, as detailed below:
- 3.1 The description of the experimental setup in Figure 1 is inadequate. For instance, is the class-incremental learning setup applied in this toy example? How are the two types of classifiers trained and tested? Which task’s accuracy is measured in the figure?
- 3.2 Does the ARC algorithm run after training each task? If so, what is the additional time required for training and inference? Additionally, should line 9 in the pseudocode be corrected to $w \leq \gamma$?
- 3.3 The scaling temperature used in $S_i$ and the total number of tasks are both denoted as $T$. Are these referring to the same setting?

**Questions:**

1. The hyperparameters used in the reported results are not fully detailed. Were the same hyperparameters applied across different methods and datasets? It seems that $\beta$ will influence the number of test samples used for training, and there is a trade-off between the number of samples and the accuracy of the pseudo-labels. How were the hyperparameters selected?

2. In the discussion following Assumption 2, the authors state that a low ratio of $c$ and $\hat{c}$ indicates a sample is more likely misclassified as belonging to a past task. However, Figure 5 suggests that a larger corresponds to better performance, which seems contradictory. Could the authors clarify this point?

---

> ### Author Response · Authors · 2024-11-20
> **Response from authors**
>
> We thank reviewer EsC3 for the thoughtful and constructive comments. It is evident that the reviewer spent considerable time carefully reading our work and provided valuable insights and suggestions. We really appreciate the detailed feedback and the recognition of the strengths in our approach. Below, we address the specific points raised by the reviewer.
>
>
> >*While the motivation to correct misclassified past task samples is sound, the proposed Task-based Softmax Score (TSS) may be problematic. For the earlier tasks, as the temperature increases, the probability distribution flattens, which could hinder the model’s ability to recognize samples from past tasks.*
>
> Thank you for pointing this out. To clarify, the denominator of TSS does not sum over logits for all classes (as in regular softmax) but is instead restricted to the classes revealed to the model up to task $i$. For earlier tasks, this results in fewer terms in the denominator, which sharpens the probability distribution rather than flattening it. To counterbalance this sharpening effect, we employ a temperature value $T > 1$ ("accommodate the imbalanced denominator for different tasks" as stated in L344).
>
> To further illustrate the impact of the temperature value, we provide the following ablation study, which demonstrates its effect on average accuracy across different methods. As shown, incorporating the temperature parameter improves performance consistently:
>
> ||Temperature |Avg Acc on Imagenet-R Inc20|
> |----:|:---:|:---:|
> |iCarL |&check;  |67.6  |
> |iCarL |&cross;  |67.4  |
> |Memo  |&check;    |68.2  |
> |Memo  |&cross;    |67.8  |
> |L2P  |&check; |75.1 |
> |L2P  |&cross;  |74.6 |
> |DualPrompt |&check; |70.5|
> |DualPrompt |&cross;  |69.9 |
>
> >*Although the experiments demonstrate the effectiveness of ARC, the paper lacks comparisons with other test-time adaptation (TTA) benchmarks, such as CoTTA.*
>
> We would like to point out that we **have compared ARC with other TTA benchmarks**, as shown in Table 3, where TENT is included as a representative TTA method. This has also been acknowledged by Reviewer hK7E.
>
> Regarding CoTTA, we tried to include it in our experiments. However, our findings reveal that CoTTA is not compatible with prompt-based methods. Directly applying CoTTA to these methods leads to a decrease in performance, as shown below:
>
> ||Avg Acc on Imagenet-R Inc20|
> |----:|:---:|
> |L2P   |72.6   |
> |L2P + CoTTA   |72.2   |
> |DualPrompt  | 69.1 |
> |DualPrompt + CoTTA   |67.7   |
>
> For non-prompt-based methods, while CoTTA does provide some improvements, the gains are less significant than those achieved by ARC. This is evident in the following comparison:
>
> ||Avg Acc on Imagenet-R Inc20|
> |----:|:---|
> |iCarL + ARC   |67.6  |
> |iCarL + CoTTA   |64.7  |
> |Memo + ARC |68.2 |
> |Memo + CoTTA |67.6 |
>
> Therefore, we did not report the comparison with CoTTA.
>
> >*Additionally, the introduction of extra data for classifier training (referred to as retention) raises concerns about increased training costs that should be addressed.*
>
> We would like to clarify that ARC **does not introduce any additional training costs**. The retention mechanism is applied solely during inference, where the classifier is adaptively rebalanced for incoming test samples. This process involves only one gradient update per sample, performed in an online manner. As a result, the computational overhead introduced by ARC is confined to the inference stage (details in the answer below), without impacting the training process.
>
> >*The description of the experimental setup in Figure 1 is inadequate. For instance, is the class-incremental learning setup applied in this toy example? How are the two types of classifiers trained and tested? Which task’s accuracy is measured in the figure?*
>
> We apologize for the confusion. To clarify:
>
> - The independent classifier in Figure 1 is trained and tested under the TIL setting, while the shared classifier is trained and tested under the CIL setting.
> - $A_i$ in the figure refers to the average accuracy of task $i$, as defined in Equation 4 of the paper.
>
> In fact, the purpose of Figure 1 is to demonstrate why the CIL problem is significantly harder than the TIL problem, i.e., primarily due to classifier bias. This highlights how the way classifiers are constructed can have a much greater impact on overall continual learning performance compared to the backbone architecture, which is a major motivation of our approach. We included this figure to bring attention to a key point that might not be immediately obvious to readers less familiar with continual learning.

---

> ### Author Response · Authors · 2024-11-20
> **Response from authors**
>
> >*Does the ARC algorithm run after training each task? If so, what is the additional time required for training and inference?*
>
> Thank you for your question. The ARC algorithm is applied after training on each task. However, we emphasize again that ARC does not interfere with the training process itself, so the additional training time required by ARC is zero.
>
> During inference, due to time limitation, we provide the extra time required for ARC on 4 representative methods (same ones we used for hyperparameter ablation)
>
> ||Additional time required|Performance Gain|
> |----:|:---:|:---:|
> |iCarL   |11%   | 6.6|
> |Memo   |8%   |2.0 |
> |L2P  | 37% |2.5|
> |DualPrompt   |32%   |1.4|
>
> As shown, there is a certain amount of additional inference time required for ARC. This is expected, as ARC is specifically designed to address classification bias during inference, which naturally incurs some additional computational cost.
>
> Furthermore, it is worth noting that research in other areas, such as large language models (LLMs), are increasingly exploring methods that enhance performance by utilizing additional inference-time computation. There is a growing consensus that additional inference time is a reasonable trade-off when it leads to significant improvements in model performance. Similarly, we believe the computational cost of our method is well-justified by the substantial performance gains it delivers.
>
>
> >*Line 9 in the pseudocode*
>
> Thank you for pointing out this typo! We will correct it in the revised version.
>
> >*The scaling temperature used in $S_i$ and the total number of tasks are both denoted as $T$. Are these referring to the same setting*
>
> Thank you again for pointing this out! These are supposed to be different notations. We will revise accordingly.
>
> >*The hyperparameters used in the reported results are not fully detailed. Were the same hyperparameters applied across different methods and datasets? It seems that $\beta$ will influence the number of test samples used for training, and there is a trade-off between the number of samples and the accuracy of the pseudo-labels. How were the hyperparameters selected?*
>
> We conducted experiments (shown in Figure 5) to demonstrate how the choice of hyperparemeters influence our method, and results show that:
>  - our method is robust to the choice of hyperparameters, a point also acknowledged by Reviewer 1cF7 and hK7E.
>  - using the same hyperparameters across all datasets and methods would still yield performance gains
>
> However, we choose not to use the same hyperparameter following previous methods. To ensure a fair evaluation, we used the best-performing values of $\gamma$ and $\beta$ for each method and dataset.
>
> >*In the discussion following Assumption 2, the authors state that a low ratio of $c$ and $\hat{c}$ indicates a sample is more likely misclassified as belonging to a past task. However, Figure 5 suggests that a larger corresponds to better performance, which seems contradictory. Could the authors clarify this point?*
>
> Thank you for your comment. We do not believe the results in Figure 5 are contradictory. The results in Figure 5 demonstrate how low the threshold should be for optimal performance. It would only be contradictory if ARC performed better when the threshold was **larger than 1**, which is not the case.
>
> To further illustrate this, we have conducted additional experiments to evaluate the performance of ARC when $\gamma = 1.1$. The results are as follows:
> ||Avg Acc on Imagenet-R Inc20|
> |----:|:---:|
> |iCarL    |65.5 (-2.1)   |
> |Memo   |67.5 (-0.7)   |
> |L2P  | 74.7 (-0.4) |
> |DualPrompt  |70.3 (-0.2)   |

---

> > ### Comment · Reviewer_EsC3 · 2024-11-25
> >
> > Thank you for your detailed and clear response, which has addressed most of my concerns. However, I still have the following questions and suggestions:
> > 1. I now understand that the temperature $𝑇$ in Task-based Softmax Score (TSS) is used to alleviate the issue of probability sharpening for earlier tasks caused by the smaller denominator. The use of $i$ in the original paper by TSS makes this formula somewhat difficult to follow.
> >
> > 2. Regarding the additional training costs, I apologize for not expressing my concerns clearly earlier. I understand that the ARC algorithm is applied after training on each task. However, I am confused about whether the model trained with ARC is used as the starting point for the next task. If this is the case, then this overhead should be accounted for when considering the training cost across all $T$ tasks. I think the authors need to provide a clearer explanation of this aspect of the method.
> >
> > 3. Thank you for providing the additional experiments, including CoTTA and the analysis of additional time required. I believe these experiments could be included in the appendix to provide a more comprehensive understanding of ARC. Additionally, regarding the additional time required, is the reported percentage relative to the original inference time? This should be clarified.
> >
> > 4. To ensure the reproducibility of results, I suggest providing a clearer table of the final hyperparameter choices across different methods and datasets.
> >
> > Overall, I think leveraging confidence scores to detect past task samples and using them to adjust the model in CIL is a reasonable approach. The authors have provided extensive experiments to demonstrate its effectiveness. Based on this, I am willing to raise my score to 5. I strongly recommend that the authors revise unclear statements in the paper, particularly in the methodology section, to improve the clarity and accessibility of the work.

---

> > > ### Author Response · Authors · 2024-11-25
> > > **Response from authors**
> > >
> > > >*However, I am confused about whether the model trained with ARC is used as the starting point for the next task. If this is the case, then this overhead should be accounted for when considering the training cost across all tasks. I think the authors need to provide a clearer explanation of this aspect of the method.*
> > >
> > > Yes, this is indeed how ARC is used. However, we respectfully disagree that the overhead should be accounted for across all tasks. In real-world CL use cases:
> > >
> > > 1. A model is initially trained on a dataset and deployed in practice.
> > > 2. Over time, new requirements emerge or new data becomes available, necessitating updates to the model.
> > > 3. The updated model is redeployed and continues to be used in practice.
> > > 4. This process repeats as additional data or requirements arise.
> > >
> > > In this workflow, what matters most is the additional computational cost incurred during Step 3—when the model is redeployed after being updated. Evaluating overhead across previous cycles seems less relevant, as the focus should be on the efficiency of deployment after updates rather than cumulative overhead from prior tasks.
> > >
> > > That said, we appreciate your perspective and will clarify this point further to ensure the process is well-understood. Thank you for bringing this up!
> > >
> > > >*Is the reported percentage relative to the original inference time?*
> > >
> > > Yes, we will clarify this.
> > >
> > > >*Reproducibility of results*
> > >
> > > Thank you for the suggestion! We will include it in the appendix. We also promise to open source the code upon acceptance.
> > >
> > > >*I strongly recommend that the authors revise unclear statements in the paper, particularly in the methodology section, to improve the clarity and accessibility of the work.*
> > >
> > > Thank you for pointing this out! We will carefully revise and clarify all statements identified as unclear to ensure the paper is as accessible as possible.
> > >
> > > We’re also curious to know if our responses have sufficiently addressed your questions. Additionally, are there are other statements you still find unclear? We would greatly appreciate your feedback so we can address them comprehensively.

---

> > > ### Author Response · Authors · 2024-12-02
> > > **Response from authors**
> > >
> > > Dear reviewer EsC3, as the end of the rebuttal period is approaching, we want to kindly check if our responses have addressed your concerns. If there are any remaining points or statements that you still find unclear, we would be happy to provide further clarification.

---

> ### Author Response · Authors · 2024-11-24
> **Response from authors**
>
> Dear reviewer, we would be grateful if you could confirm whether our response has addressed your concerns. Please do not hesitate to let us know whether there is anything else you would like to see clarified or improved before the end of the rebuttal period.

---

### Official Review · Reviewer_ALn4 · 2024-11-04

**Soundness:** 4
**Presentation:** 4
**Contribution:** 4
**Rating:** 8
**Confidence:** 4

**Summary:**

The authors observe that the main issue of forgetting is not in model itself but the classifier layer. Base on such observation, they introduce a way to balance the calssifying layer between current task and previous task. Instead of naively mask the current task output, the authors argue that it is not suitable since the balance between previous class is also imbalance. They come up with a new algorithm to tackle the problem by two assumptions, retention and correction. With the new algorithm ARC, it gain the performance on various datasets and methods.

**Strengths:**

1. The observation of forgetting majorly happen in classifer sounds reasonable and inspring. The proposed method, ARC, also solve it well.
2. ARC can be generally applied to different methods is a big advantage that it can plug and play easily with existing countinual learning techniques.

**Weaknesses:**

1. Need to report the training time during inference, since it might add overhead when deployed to the real-world.

**Questions:**

1. What if the incoming sample from the past task is not allow to use during inference? For example, it is deployed to the embbedding device that need real-time inference, will additional training add much overhead? Or is it possible to not update at every step? It will be interesting to provide such results.
2. Can this approach extend to other computer vision task such as detection and segmentation? It would be great if the methods can also fix the problem in such scenario by re-balancing the classifcation head in object detection.

Minor Correction
1. At Table 1, it seems strange that for CodaPrompt, the position of the difference for Split CIFAR-100 Inc5 is oppositve. The difference sign should be 5.8 $\textcolor{red}{(+0.1)}$ instead of 5.7 $\textcolor{green}{(-0.1)}$.

---

> ### Author Response · Authors · 2024-11-20
> **Response from authors**
>
> We thank reviewer ALn4 for the thoughtful and constructive comments. It is evident that the reviewer spent considerable time carefully reading our work and provided valuable insights and suggestions. We really appreciate the detailed feedback and the recognition of the strengths in our approach. Below, we address the specific points raised by the reviewer.
>
> >*Need to report the training time during inference, since it might add overhead when deployed to the real-world.*
>
> Thanks for the advice! Due to time limitation, we provide the extra time required for ARC on 4 representative methods (same ones we used for hyperparameter ablation):
>
> ||Additional time required|Performance Gain|
> |----:|:---:|:---:|
> |iCarL   |11%   | 6.6|
> |Memo   |8%   |2.0 |
> |L2P  | 37% |2.5|
> |DualPrompt   |32%   |1.4|
>
> As shown, there is a certain amount of additional inference time required for ARC. This is expected, as ARC is specifically designed to address classification bias during inference, which naturally incurs some additional computational cost.
>
> Furthermore, it is worth noting that research in other areas, such as large language models (LLMs), are increasingly exploring methods that improve performance by utilizing additional inference-time computation. There is a growing consensus that additional inference time is a reasonable trade-off when it leads to significant improvements in model performance. Similarly, we believe the computational cost of our method is well-justified by the substantial performance gains it delivers.
>
> >*What if the incoming sample from the past task is not allow to use during inference? For example, it is deployed to the embbedding device that need real-time inference, will additional training add much overhead? Or is it possible to not update at every step? It will be interesting to provide such results.*
>
> Thank you for raising this question. We would like to emphasize again that our method does not store any test samples in memory. Instead, it operates using on-the-fly test samples and performs a single gradient update as each sample is processed by the model. So whether or not past task sample is allowed to use is irrelevant to our method.
>
> Regarding the computational overhead, as noted in our response above, the additional time required for inference varies across methods. In the worst-case scenario with L2P, our method results in a frame-per-second (FPS) drop from approximately 60 to 43 on an NVIDIA 4090 GPU using the ViT-B/16 backbone, which still meets the criteria for real-time inference.
>
> Reducing the frequency of updates is indeed a promising approach to further minimize the additional inference time. For instance, this could involve quantifying how "biased" the classifier is based solely on a batch of test samples before deciding whether an update is necessary. However, exploring this idea in detail falls outside the scope of this rebuttal. We appreciate your suggestion and consider it an interesting direction for future work.
>
> >*Can this approach extend to other computer vision task such as detection and segmentation? It would be great if the methods can also fix the problem in such scenario by re-balancing the classifcation head in object detection.*
>
> Yes, we believe that the idea of ARC can be applied to any computer vision task where classifier bias is a challenge. In fact, there are already works exploring similar directions, such as [1], which addresses bias in continual semantic segmentation. However, applying ARC to tasks like detection or segmentation would require certain modifications to the implementation details to account for the unique characteristics of these tasks. For example, in object detection, ARC would need to re-balance not only the classification head but also adapt to region proposals or bounding box predictions to mitigate potential biases.
>
> [1] RBC: Rectifying the Biased Context in Continual Semantic Segmentation
>
> >*At Table 1, it seems strange that for CodaPrompt, the position of the difference for Split CIFAR-100 Inc5 is oppositve. The difference sign should be 5.8 (+0.1) instead of 5.7 (-0.1)*
>
> Thank you for pointing this out! We agree that your suggested revision would indeed improve the clarity of the table. We will revise accordingly.

---

> > ### Comment · Reviewer_ALn4 · 2024-11-24
> >
> > Thanks for clarifying the concerns. It is good to know that ARC does not need much computational power when deploying. I believe test-time training is a promising approach to address continual learning challenges, especially the cost is reasonable. The authors should provide a more detailed discussion of this in the related work section, emphasizing that the use of additional computational power is generally acceptable. Overall, I find ARC to be a promising method, and I am considering increasing the score.

---

> > > ### Author Response · Authors · 2024-11-25
> > > **Response from authors**
> > >
> > > Thank you for taking the time to revisit our work and for your encouraging feedback! We’re glad to hear that the computational efficiency of ARC aligns with your expectations. We appreciate your suggestion to provide a more detailed discussion on TTA in the related work section, and we will ensure it is addressed in our final revision to offer greater clarity and context.

---

> > > ### Author Response · Authors · 2024-12-02
> > > **Follow-Up on Review Feedback**
> > >
> > > Dear reviewer ALn4, we noticed that you mentioned the possibility of raising your score based on the discussion and our rebuttal. If there are any additional points or clarifications you would like us to address to support this process, we'd be happy to provide further details. We appreciate your time and effort in evaluating our work and look forward to your decision.

---

> ### Author Response · Authors · 2024-11-24
> **Response from authors**
>
> Dear reviewer, we would be grateful if you could confirm whether our response has addressed your concerns. Please do not hesitate to let us know whether there is anything else you would like to see clarified or improved before the end of the rebuttal period.

---

### Official Review · Reviewer_1cF7 · 2024-11-04

**Soundness:** 2
**Presentation:** 4
**Contribution:** 3
**Rating:** 6
**Confidence:** 4

**Summary:**

The paper presents a novel approach to class-incremental memory-free continual learning by integrating out-of-distribution detection with model self-correction. Specifically, the authors observe that performance degradation in continual learning largely stems from the classifier head and suggest adapting it on test samples using an entropy minimization loss while refining its predictions through a modified softmax score. Experimental results on Split CIFAR and Split ImageNet-R demonstrate the method's robustness and effectiveness across both replay-based and memory-free continual learning techniques.

**Strengths:**

The submission offers a comprehensive analysis of the dynamics of catastrophic forgetting, with a focus on recency bias, and proposes a simple and effective method to alleviate it that can be combined with any continual learning method.

The experimental evaluation is rigorous, covering relevant baselines, standard datasets, and appropriate ablations. The new approach demonstrates robustness to method-specific hyperparameter choices and provides a performance boost across continual learning algorithms.

The paper is well-written and structured, with clearly presented arguments, consistent notation, and high-quality figures that effectively convey the main points.

**Weaknesses:**

The main weakness of the paper is that it makes assumptions about the continual learning setting that are not well justified. Computational constraints are often more critical than memory limitations (see Prabhu et al. 2023, *Computationally Budgeted Continual Learning: What Does Matter* and Roth et al. 2024, *A Practitioner's Guide to Continual Multimodal Pretraining*). A significant limitation of the proposed method is its requirement for additional gradient updates during inference, even if only a single update per sample is needed.

Additionally, performing a hyperparameter search for the adaptive self-retention procedure could lead to overfitting on the test samples.

To improve the evaluation, it would be best to: a) quantify the computational cost of the self-retention procedure and b) ensure hyperparameter tuning is conducted on a separate dataset from the evaluation dataset.

**Questions:**

Did you run the hyperparameter search for the adaptive self-retention procedure on the same dataset that was used for evaluation?

Have you tried applying your method to other continual learning algorithms, like BEEF, EASE, MEMO, and FOSTER?

In line 88, you formulate the two observations as empirical findings. However, in Section 3.3, they are called assumptions. Do you have any empirical results that show to what extent these actually hold?

---

> ### Author Response · Authors · 2024-11-20
> **Response from authors**
>
> We thank reviewer 1cF7 for the thoughtful and constructive comments. It is evident that the reviewer spent considerable time carefully reading our work and provided valuable insights and suggestions. We really appreciate the detailed feedback and the recognition of the strengths in our approach. Below, we address the specific points raised by the reviewer.
>
>
> >*Quantify the computational cost of the self-retention procedure*
>
> Thanks for the advice! Due to time limitation, we provide the extra time required for ARC on 4 representative methods (same ones we used for hyperparameter ablation) in the table below:
>
> ||Additional time required|Performance Gain|
> |----:|:---:|:---:|
> |iCarL   |11%   | 6.6|
> |Memo   |8%   |2.0 |
> |L2P  | 37% |2.5|
> |DualPrompt   |32%   |1.4|
>
> As shown, there is a certain amount of additional inference time required for ARC. This is expected, as ARC is specifically designed to address classification bias during inference, which naturally incurs some additional computational cost.
>
> Furthermore, it is worth noting that research in other areas, such as large language models (LLMs), are increasingly exploring methods that improve performance by utilizing additional inference-time computation. There is a growing consensus that additional inference time is a reasonable trade-off when it leads to significant improvements in model performance. Similarly, we believe the computational cost of our method is well-justified by the substantial performance gains it delivers.
>
> >*Did you run the hyperparameter search for the adaptive self-retention procedure on the same dataset that was used for evaluation? Better to ensure hyperparameter tuning is conducted on a separate dataset from the evaluation dataset.*
>
> Yes, the hyperparameter search is ran on the same dataset, following previous methods. Furthermore, our experiments in Fig 5 shows that our method is robust to the choice of hyperparameters, and that even using the same hyperparameter across all datasets and methods would still yield performance gains. Therefore, to ensure a fair comparison, we used the best-performing values of $\gamma$ and $\beta$ for each method and dataset.
>
>
> >*Have you tried applying your method to other continual learning algorithms, like BEEF, EASE, MEMO, and FOSTER?*
>
> Thank you for your question. We actually have applied ARC to MEMO, and the results are reported in Table 1 of the paper. Additionally, ARC has been applied to a total of eight methods, covering a diverse range of approaches, including prompt-based and non-prompt-based methods as well as rehearsal-based and non-rehearsal-based methods. We believe this already demonstrates the versatility and generalizability of our method.
>
> Due to time constraints, it may be challenging to provide results for additional methods, such as BEEF, EASE, and FOSTER, within the scope of this submission. However, we consider this an interesting direction for future exploration.
>
> >*In line 88, you formulate the two observations as empirical findings. However, in Section 3.3, they are called assumptions. Do you have any empirical results that show to what extent these actually hold?*
>
> Yes, the empirical validation of assumptions 1 and 2 is included in Table 4 and 5 of the paper.

---

> > ### Comment · Reviewer_1cF7 · 2024-11-22
> > **Response to the authors**
> >
> > Thank you for responding to my concerns and quantifying the extra time required at inference.
> >
> > Just to be clear, I did not expect you to run any additional experiments, I was just curious if you applied ARC on top of more recent methods. Thanks for pointing out MEMO, I missed it in Table 1.
> >
> > ARC is well motivated, robust, and seems to lead to improvements on top of a wide range of continual merging methods. While I acknowledge the argument about test-time compute in LLMs, whether the accuracy gain is worth a potential 8-37% hit in latency will depend on the particular application. To make a stronger case for the method, I would recommend testing how it scales with the number of tasks in the training sequence.
> >
> > Thank you again for your response. I will maintain my original score.

---

> > > ### Author Response · Authors · 2024-11-23
> > > **Response from authors**
> > >
> > > We thank you for your recognition of our work and for taking the time to provide thoughtful feedback. Your insights and suggestions are greatly appreciated!

---

### Meta-Review · Area_Chair_aEgt · 2024-12-19

**Metareview:**

The paper introduces an Out-of-Task Detection method, OTD, designed to accurately identify samples from past tasks during testing, aiming to overcome classification layer’s bias towards the most recent task in memory-free continual learning.

**Strengths**
- The idea of adaptively adjusting and correcting model predictions using test samples is interesting and well-suited to continual learning.
- The proposed approach achieves good performance.

**Weaknesses**
- Lack sufficient details of hyperparamter tuning and the training process of the model.
- Lack a detailed comparison with existing test-time adaptation approaches.

Overall, leveraging test samples to address catastrophic forgetting appears to be a new and effective approach, particularly in the context of memory-free continual learning.

**Additional Comments On Reviewer Discussion:**

Multiple reviewers engaged in an intensive discussion with the authors during the rebuttal process. The authors conducted additional experiments and provided detailed responses, which significantly improved the quality and clarity of the paper.

---

> ### Public Comment · ~Haoran_Chen4 · 2025-02-18
> **Refinement of paper title**
>
> Dear AC,
>
> We thank all reviewers and AC for the insightful review and the constructive feedback provided. We greatly appreciate the positive recognition of our work, particularly the effectiveness of leveraging test samples to address catastrophic forgetting for memory-free continual learning. Based on the feedback, we are considering refining the title of our paper to better reflect its contribution, specifically 'Adaptive Retention & Correction: Test-Time Training for Continual Learning.
>
> However, we are not sure how we can update the title on the OpenReview page. Could you let us know how to do that?
>
> Best,
> Authors

---

> > ### Comment · Area_Chair_aEgt · 2025-02-18
> >
> > Dear Authors,
> >
> > I believe you may directly change the title on OpenReview. Please refer to the following part of the ICLR 2025 Camera Ready Instructions:
> >
> > >**Title, Abstract, Supplemental, etc:** In addition to uploading the final PDF for the paper, you may change the paper title, abstract, keywords, tldr, primary area, and supplemental PDF. Changes to the title should be in response to the reviewer or area chair comments, and must not significantly change the scope of the paper. All these changes can be made directly on OpenReview.
> >
> > Best,
> >
> > AC

---

### Decision · Program_Chairs · 2025-01-22

Accept (Poster)